# Visual Detection of Dopamine with CdS/ZnS Quantum Dots Bearing by ZIF-8 and Nanofiber Membranes

**DOI:** 10.3390/ijms251910346

**Published:** 2024-09-26

**Authors:** Jiadong Hu, Jiaxin Li, Qunqun Guo, Guicai Du, Changming Li, Ronggui Li, Rong Zhou, Hongwei He

**Affiliations:** 1Industrial Research Institute of Nonwovens & Technical Textiles, College of Textiles & Clothing, Qingdao University, Qingdao 266071, China; bhujiadong@163.com (J.H.); rzhou@qdu.edu.cn (R.Z.); 2College of Life Sciences, Qingdao University, Qingdao 266071, China; 2021023499@qdu.edu.cn (J.L.); gqunqun@163.com (Q.G.); duguicai@qdu.edu.cn (G.D.); lrg@qdu.edu.cn (R.L.); 3Schneider Institute of Industrial Technology, Qingdao University, Qingdao 266071, China

**Keywords:** dopamine detection, CdS/ZnS quantum dots, electrospinning, ZIF-8, high sensitivity

## Abstract

Dopamine (DA) is a widely present, calcium cholinergic neurotransmitter in the body, playing important roles in the central nervous system and cardiovascular system. Developing fast and sensitive DA detection methods is of great significance. Fluorescence-based methods have attracted much attention due to their advantages of easy operation, a fast response speed, and high sensitivity. This study prepared hydrophilic and high-performance CdS/ZnS quantum dots (QDs) for DA detection. The waterborne CdS/ZnS QDs were synthesized in one step using the amphiphilic polymer PEI-g-C14, obtained by grafting tetradecane (C14) to polyethyleneimine (PEI), as a template. The polyacrylonitrile nanofiber membrane (PAN-NFM) was prepared by electrospinning (e-spinning), and a metal organic frame (ZIF-8) was deposited in situ on the surface of the PAN-NFM. The CdS/ZnS QDs were loaded onto this substrate (ZIF-8@PAN-NFM). The results showed that after the deposition of ZIF-8, the water contact angle of the hydrophobic PAN-NFM decreased to within 40°. The nanofiber membrane loaded with QDs also exhibited significant changes in fluorescence in the presence of DA at different concentrations, which could be applied as a fast detection method of DA with high sensitivity. Meanwhile, the fluorescence on this PAN-NFM could be visually observed as it transitioned from a blue-green color to colorless, making it suitable for the real-time detection of DA.

## 1. Introduction

Dopamine (DA) exists in neural tissues and body fluids and plays an important role in the central nervous system and cardiovascular system [1,2,3]. Abnormal concentration of DA in the human body can lead to diseases such as Parkinson’s disease [4,5], schizophrenia [6,7], Alzheimer’s disease [8], and addiction disorders [9]. The design and preparation of a highly selective and sensitive biosensor platform for detecting ultra-low levels of DA is of great significance in disease management, monitoring, and treatment [10]. The commonly used detection methods for DA include microdialysis [11], fluorescence spectroscopy [12,13,14,15,16], electrochemical methods [17,18,19,20], colorimetric methods [21,22], and high-performance liquid chromatography [23,24]. The fluorescence probe, as a detection method, has the advantages of high sensitivity, absence of large instruments, simple operation, and a low cost, and has ideal application prospects for the detection of DA.

Quantum dots (QDs), a new type of nanocrystal, have a series of advantages such as a long lifespan, high fluorescence emission, high photo stability, and surface modifiability, which are the reasons they can be used as biosensors and technology for fluorescence imaging to detect biomolecules [25]. There are various types and preparation methods of QDs [26,27,28,29,30]. Some inorganic compounds, such as the metal sulfide series QDs, have been widely used due to their simple preparation and wide excitation and emission spectrum range [31,32,33]. These kinds of QDs, prepared from oil-based systems, are often difficult to apply to the detection of organisms and biomolecules, and the synthesis of such quantum dots in aqueous systems has become a research focus. Metal organic frameworks (MOFs) are a type of material with high porosity, highly specific surface area, and low density. The zeolitic imidazolate framework-8 (ZIF-8) is the most representative MOF material. High porosity MOFs provide a platform for loading QDs to prevent their aggregation, and the binding of QDs to MOFs exhibits good dispersibility and stability [34,35]. Li et al. [36] constructed a ratio-type fluorescent probe for the rapid detection of 2,6-pyridinedicarboxylic acid through the in situ doping of rhodamine 6 G and CdS QDs into ZIF-8.

Electrospinning (e-spinning) is a technique that has been developed to prepare nanofiber membranes (NFM) with high porosity, a large specific surface area, an adjustable structure, and easy surface modification. Zhang et al. [37] grew ZIF-8 loaded with carbon dots (E-CDs) in situ onto biomass fiber membranes (PCL/PLA) and constructed a colorimetric solid-state fluorescence detection platform with high sensitivity and stability for detecting trace amounts of Cu^2+^. Lee et al. [38] successfully prepared side-by-side nanofiber membranes of europium metal organic frameworks (Eu-MOFs) and terbium metal organic frameworks (Tb-MOFs) using the e-spinning method. Adjusting the ratio of Eu-MOFs and Tb-MOFs emitting red and green light on both sides can result in a luminescent sensor film emitting yellow light.

In this article, based on our previous work on the synthesis of waterborne CdS/ZnS QDs with amphiphilic polymers of PEI-g-C14 as templates [39], we generated electrospun (e-spun) polyacrylonitrile (PAN) nanofiber membranes which were then deposited in situ with the MOF (ZIF-8) and loaded with the CdS/ZnS QDs. The PAN-NFM bearing the QDs/ZIF-8 had a high fluorescence performance and sensitivity for DA detection, achieving relevancy for the application of fluorescent fiber membranes in DA visualization and detection. This strategy provides a nanofiber membrane for wound dressing which can also detect DA visually.

## 2. Results

### 2.1. Morphology and Structures of the Composite PAN-NFM

The morphologies of the PAN-NFM and its composites loaded with ZIF-8 and QDs were characterized by SEM. The pristine PAN-NFM was a nonwoven structure formed by uniform nanofibers and had an average diameter of 231 nm (Figure 1a–c). The morphologies of the ZIF-8@PAN-NFM and the QDs/ZIF-8@PAN-NFM were shown in Figure 1 as well. There were many obvious and irregular particles (30–40 nm) attached to the surface of the fibers (Figure 1d,e,g,h), and their average diameters were 292 nm and 304 nm, respectively (Figure 1f,i). Compared to the pristine PAN-NFM (Figure 1c), the average diameters of the ZIF-8@PAN-NFM and the QDs/ZIF-8@PAN-NFM were increased. The deposition of much amount of ZIF-8 resulted in increasing average fiber of ZIF-8@PAN-NFM and QDs loading did not increase obviously average fiber of QDs/ZIF-8@PAN-NFM, which could be attributed to QDs being mainly located in the pores of the ZIF-8.

The chemical structures of the PAN-NFM, ZIF-8@PAN-NFM, QDs, and QDs/ZIF-8@PAN-NFM were characterized by FT-IR, and are shown in Figure 2. The characteristic absorption peaks at 2242 cm^−1^ and 2929 cm^−1^ were assigned to the stretching vibration absorption of CN and CH_2_ of the PAN’s chain, which appeared in all spectra in Figure 2. However, an obvious decrease in the peaks occurred in the spectra of the ZIF-8@PAN-NFM and QDs/ZIF-8@PAN-NFM due to ZIF-8 loading. The peaks at 1590 cm^−1^ and 1145 cm^−1^ were attributed to the stretching vibration of C=N and C–N, respectively, which were caused by the structure of ZIF-8. The peak at 758 cm^−1^ was attributed to the absorption of the out-of-plane bending vibration of C-H, originated from imidazole for synthesizing ZIF-8. The CdS/ZnS QDs did not appear significantly in the spectrum of the QDs/ZIF-8@PAN-NFM, which could be because they are inorganic substances and of low content.

As shown in Figure 3, the PAN-NFM started decomposing at 300 °C, and due to cyclization reactions and carbonization at high temperatures, the weight loss was about 43% when the temperature reached 600 °C. As the inorganic component, CdS/ZnS QDs lost a total weight of about 27% because the sulfides were not very stable. ZIF-8 is also a type of MOF that is not resistant to high temperatures. However, the ZIF-8 and QDs loaded on the PAN-NFM may have caused more difficulties in the PAN’s cyclization reactions and carbonization, as the weight losses of the ZIF-8@PAN-NFM and QDs/ZIF-8@PAN-NFM were 49% and 59% at 600 °C, respectively, which were higher than that of the pristine PAN-NFM.

### 2.2. Wettability

As shown in Figure 4, the contact angle of the PAN-NFM was 80°, and that of the ZIF-8@PAN-NFM was 65°, which showed that its wettability was improved. The contact angle of the QDs/ZIF-8@PAN-NFM was as low as 39°, and it was observed that the testing water drop could quickly infiltrate it. 

### 2.3. Pore Size Distribution

As a porous e-spun nonwoven membrane, the pore size and porosity affected its performance. The pore size distribution and porosity of the prepared PAN-NFM and its composites with ZIF-8 and QDs are shown in Figure 5. The pore size of the PAN-NFM was around 1.07–1.21 μm, with a pore size of 1.16 μm accounting for 62.0% of the total material. After growing ZIF-8 in situ, the pore size of the ZIF-8@PAN-NFM decreased, ranging from 1.0–1.15 μm, with a pore size of 1.08 μm accounting for 50.0% of the total material. High porosity facilitated the diffusion of the QD solution to ZIF-8, improving the response rate. After adsorbing the QDs, the pore size of the QDs/ZIF-8@PAN-NFM was 0.86–1.01 μm, possibly attributed to soaking the ZIF-8@PAN-NFM in the QD solution, which reduced the pores of the fibers.

### 2.4. Fluorescence of the QDs/ZIF-8@PAN-NFM

The fluorescence properties of the prepared CdS/ZnS QDs and their aqueous solutions, as well as their selectivity and sensitivity to DA, have been investigated [39]. The fluorescence performance in the fiber membrane (QDs/ZIF-8@PAN-NFM), compared to the QD solution and the ZIF-8@PAN-NFM, are shown in Figure 6. Compared to the fluorescence spectrum of the QD solution, there was a significant blue shift in the emission peak position, which may be due to the strong quantum confinement effect of luminescent nanomaterials transferring from the solution to the nanopore of ZIF-8, accompanied by changes in the local dielectric constant and polarity of the luminescent material environment. The fluorescence spectrum (blue “3”) of the ZIF-8@PAN-NFM without QDs showed that during the testing of the membrane sample, there was a much larger noise signal on the fluorescence spectrometer.

### 2.5. Application of the QDs/ZIF-8@PAN-NFM for Visual Test of DA

As shown in Figure 7a, under UV light, the QDs/ZIF-8@PAN-NFM emitted blue-green fluorescence, which was also exhibited when QDs were uniformly dispersed on the PAN-NFM. As the concentration of DA increased, the fluorescence intensity of the QDs/ZIF-8@PAN-NFM gradually weakened, which was characterized by the fluorescence spectrometer, shown in Figure 7b. When the concentration of DA was 10 mmol/L, the fluorescence of the QDs/ZIF-8@PAN-NFM was quenched.

The concentration of DA resulting in fluorescence changes were similar to values within the normal range of DA in the human body, which are lower than 5.0 mM. Therefore, an PAN-NFM for detecting DA can be designed. As shown in Figure 8, under UV light and with an increase of DA concentration from 0 mM to 7.5 mM, the fluorescence intensity of these strips of QDs/ZIF-8@PAN-NFM gradually decreased, showing a change from blue-green to colorless, and forms a test paper-like surface to enable a visual, semi-quantitative analysis method.

Using the QDs/ZIF-8@PAN-NFM, the detection of DA can be semi-quantitatively analyzed by naked eye observation under ultraviolet light (Figure 9). This method presented a visual, fast, and sensitive fluorescence detection method for DA. Moreover, this test paper-like surface is a solid-phase PAN-NFM, which is portable and suitable for real-time detection.

## 3. Discussion

This ZnS/CdS QDs used were characterized by means of HRTEM in our previous work [39], and the crystalline particles were approximately 3.5–6.5 nm with a lattice spacing of 0.26 nm. Therefore, some QDs could have been in the pores of the ZIF-8 loading onto the surface of the PAN nanofibers. Compared to these fibers loaded with QDs, adding the e-spinning solution with the ZIF-8@PAN-NFM as a supporter can effectively prevent the aggregation of QDs, thereby avoiding fluorescence resonance energy transfer (FRET) among QDs. Therefore, the e-spun nanofibers (membranes) loaded with QDs/ZIF-8 can be subjected to color-related detection and characterization. As shown in Figure 10, the distribution and orientation of the C, N, and Zn elements of the ZIF-8@PAN-NFM in SEM and EDS spectra are consistent with the nanofibers, and the presence of the Zn element showed the successful loading of ZIF-8 onto the fibers. Accordingly, the presence of the C, N, Zn, Cd, and S elements in Figure 10 indicated that the added CdS/ZnS QDs were successfully loaded onto the ZIF-8@PAN-NFM. The FT-IR and TG results also indicated that there obtained the composite of PAN-NFM loaded with ZIF-8 and QDs.

The PAN e-spun membrane has been widely used. Its molecular chain and non-woven nanostructure lead to insufficient hydrophilicity which, in turn, affects its applications in medical and health care, biomedicine, and other fields. Surface modification of the e-spun membrane could improve its wettability [40,41,42]. The low contact angle of the hydrophilic QDs/ZIF-8@PAN-NFM and its good air permeability based on the pore analysis results showed this membrane could be promising application in the above fields.

At the same time, the QDs/ZIF-8@PAN-NFM serves as a fluorescent carrier that facilitates the diffusion of the DA solution, resulting in uniform fluorescence for visual DA testing.

## 4. Materials and Methods

### 4.1. Materials

PAN (Molecular weight: 80,000), dopamine, sodium hydroxide (NaOH), and hydrochloric acid (HCl) were purchased from the Aladdin Biochemical Technology Co., Ltd. (Shanghai, China). N,N-dimethylformamide (DMF), dimethylimidazole, ethanol, methanol, Zn(NO_3_)_2_·6H_2_O, CdCl_2_·2.5H_2_O, Zn_2_SO_4_·7H_2_O, Na_2_S·9H_2_O, and NaCl were all obtained from the Macklin reagent Co., Ltd. (Shanghai, China). All of the chemicals were of analytical grade and applied without further purification.

### 4.2. Experimental Section

#### 4.2.1. Preparation of the PAN-NFM

A total of 1.0 g of PAN and 9 mL of DMF were mixed in 20 mL bottles equipped with a magnetic stirrer. After stirring at room temperature for 10 h and completely dissolving the PAN, 4 mL of the solution was added into a syringe. An e-spinning machine (DP30, Tianjin Yunfan Technology Co., Ltd., Tianjin, China) was applied in a controlled environment (i.e., temperature of 30 ± 2 °C, relative humidity of 50 ± 5%, a high voltage power supply of 18 kV, a 19 G spinning needle, and a feed (propulsion) rate of 0.5 mL/h). The as-spun fibers were collected in a drum at a speed of 10 r/min and a tip-to-collector distance of 20 cm. After 6 h, the nanofiber membrane (PAN-NFM) was obtained and dried in a vacuum at 60 °C to ensure the removal of residual solvents.

#### 4.2.2. In Situ Growth of ZIF-8 on the PAN-NFM Surface

A total of 0.25 g of the obtained PAN-NFM (20 cm × 28 cm) was immersed in 50 mL of NaOH solution (0.12 g/mL in water) at 75 °C and maintained for 20 min. Then, the PAN-NFM was washed three times with deionized water and ethanol, respectively, and dried at room temperature. In a 500 mL beaker, the treated PAN-NFM was immersed into a Zn(NO_3_)_2_ (120 mmol/L) solution, prepared by dissolving 1.75 g of Zn(NO_3_)_2_·6H_2_O into 50 mL of a methanol and ethanol solvent mixture (volume ratio 1:1). Half an hour later, 50 mL of 2-methylimidazole (480 mmol/L) solution, produced by dissolving 1.97 g of 2-methylimidazole into 50 mL of a methanol and ethanol solvent mixture (volume ratio 1:1), was added with a magnetic stirrer. The ZIF-8 was synthesized and growing in situ on the surface of the PAN-NFM, which was named after ZIF-8@PAN-NFM. After 2 h, the ZIF-8@PAN-NFM was taken out and washed three times with deionized water and ethanol, respectively.

#### 4.2.3. The Adsorption of CdS/ZnS QDs onto the ZIF-8@PAN-NFM for Visually Testing DA

The method to prepare waterborne CdS/ZnS QDs for the testing of DA in solution was reported in the previous work [39]. An amphiphilic polymer, PEI-g-C14, was synthesized and used as a template to fabricate CdS/ZnS QDs. The typical process is shown here: 0.5 g of home-made copolymer PEI-g-C_14_ was dissolved into 50 mL of deionized water and then mixed with 30 mL of CdCl_2_ (0.01 mol/L) aqueous solution. After 20 min, 20 mL of a Na_2_S (0.025 mol/L) aqueous solution was added dropwise to the mixed solution at 60 °C. After 0.5 h, 2.5 mL of a ZnSO_4_ (0.08 mol/L) aqueous solution was introduced in a dropwise manner. Finally, the reacted solution was cooled to room temperature and dialysis was performed to remove the low molecular weight substances that did not participate in the reaction, ultimately resulting in a solution of CdS/ZnS QDs (5 mmol/L).

The ZIF-8@PAN-NFMs were cut into 3 × 3 cm slices, and these slices immersed into the aforementioned solution of CdS/ZnS QDs for 12 h. Then, they were taken out and dried at room temperature. 100 μL of freshly prepared DA solutions with concentrations of 0, 0.1, 1, 10, and 100 mmol/L, respectively, was added dropwise on the slices. The fluorescence performance changes were characterized on a FL1000 fluorescence spectrometer and observed under a UV lamp. The preparation process of adsorbing the CdS/ZnS QDs onto the ZIF-8@PAN-NFM for visual testing of DA is illustrated in Figure 10.

### 4.3. Characterization

#### The Morphology and Properties of the ZIF-8@PAN-NFM

The as-spun PAN-NFM and the ZIF-8@PAN-NFM were observed using a scanning electron microscope (SEM, JSM-7800F, JEOL Ltd., Tokyo, Japan). ImageJ software (V1.51, NIH, Bethesda, MD, USA) was applied to calculate the diameter of the fibers from the SEM images. The chemical structures were characterized by Fourier transform infrared spectroscopy (Nicolet iS10, Thermo Fisher, Waltham, MA, USA) and an energy dispersive spectrometer (EDS, Genesis Apollo XL, AMETEK Inc., Berwyn, PA, USA). The thermal properties were recorded using an TGA/DSC 3 + instrument (Mettler-Toledo, Columbus, OH, USA) in a nitrogen atmosphere from 25 °C to 600 °C at a heating rate of 10 °C/min.

A contact angle goniometer (JY-Phb, Chengde Jinhe Co., Ltd., Chengde, China) was utilized to measure the water contact angle (WCA) for characterizing the surface properties changes of the PAN-NFM. The pore size distribution was characterized on a tester (TOPAS PSM-165, Frankfurt, Germany) using the bubbling method (the tested area was 2.01 cm^2^).

## 5. Conclusions

In this research, the PAN nanofiber membranes with excellent properties and pore structure were prepared using an e-spinning technique. A typical MOF, ZIF-8, was successfully loaded onto the PAN-NFM through an in situ deposition method, and then CdS/ZnS QDs were adsorbed onto the PAN-NFM. Through characterization of the morphology, pore size distribution, wettability, and fluorescence intensity of the QDs/ZIF-8@PAN-NFM, we demonstrated that ZIF-8 and QDs were uniformly and stably adhered to the membrane surface while maintaining its original properties. The hydrophilicity of the PAN-NFM was enhanced, and QDs could quickly interact with the ZIF-8@PAN-NFM, facilitating reactions with the biomolecule analyte, DA. Under UV light, the QDs/ZIF-8@PAN-NFM emitted blue-green fluorescence, and the fluorescence intensity gradually weakened with the increase of DA concentration until it was completely quenched. The QDs/ZIF-8@PAN-NFM could be cut similar to test paper and be semi-quantitatively analyzed by naked eye observation with fast detection and high sensitivity. Moreover, this assembly would be portable and suitable for real-time detection.

## Figures and Tables

**Figure 1 ijms-25-10346-f001:**
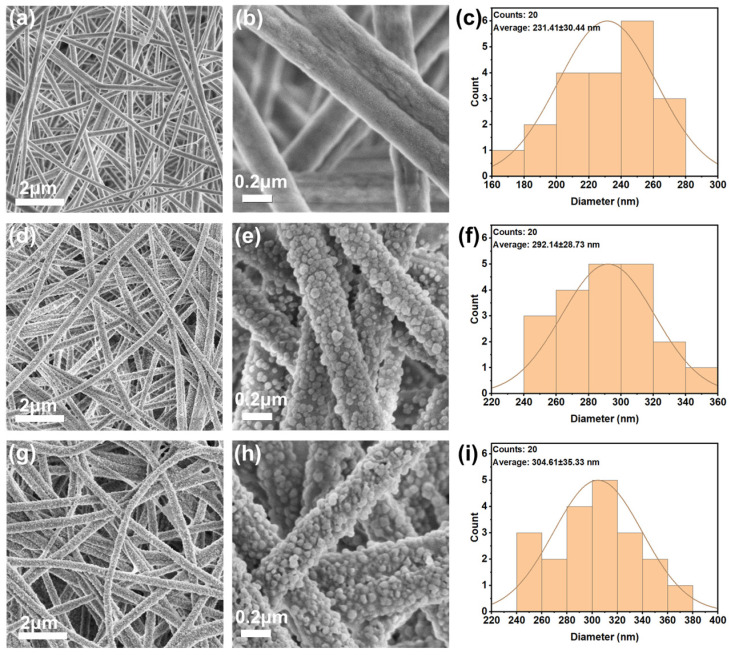
The original and enlarged SEM pictures of the following: (**a**,**b**) pristine PAN-NFM, (**d**,**e**) ZIF-8@PAN-NFM, and (**g**,**h**) QDs/ZIF-8@PAN-NFM. Their diameter distribution is shown in (**c**,**f**,**i**), respectively.

**Figure 2 ijms-25-10346-f002:**
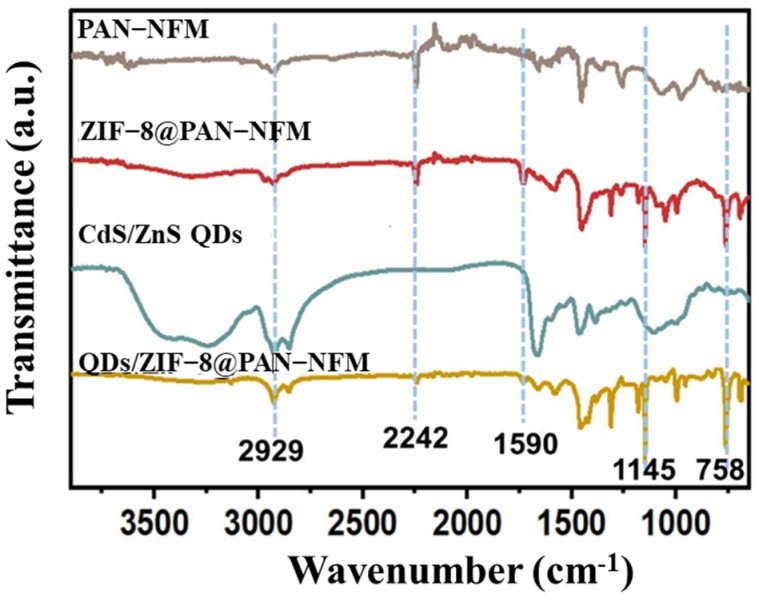
The FT-IR spectra of the pristine PAN-NFM, ZIF-8@PAN-NFM, CdS/ZnS QDs, and QDs/ZIF-8@PAN-NFM.

**Figure 3 ijms-25-10346-f003:**
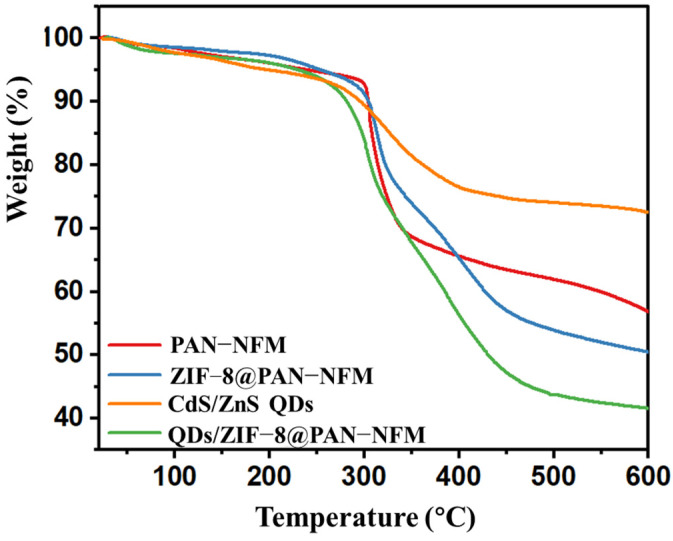
The thermal properties and weight loss percentages of the pristine PAN-NFM, ZIF-8@PAN-NFM, CdS/ZnS QDs, and QDs/ZIF-8@PAN-NFM.

**Figure 4 ijms-25-10346-f004:**
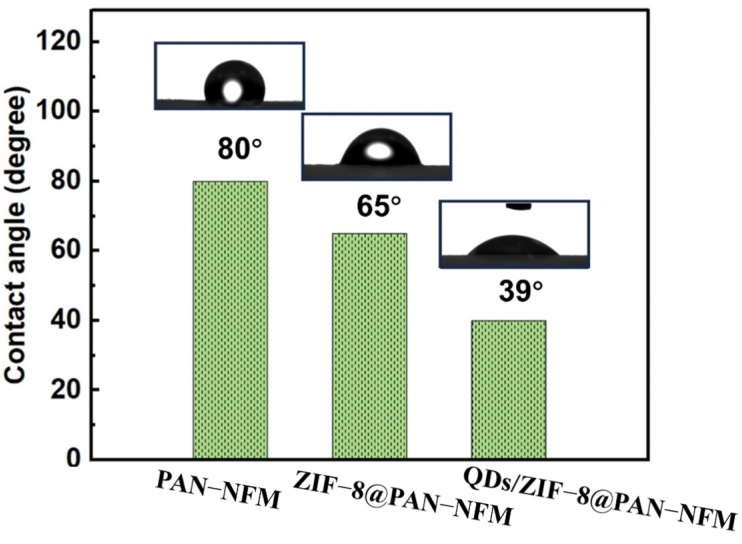
The wettability (water contact angle) of the PAN-NFM, ZIF-8@PAN-NFM, and QDs/ZIF-8@PAN-NFM.

**Figure 5 ijms-25-10346-f005:**
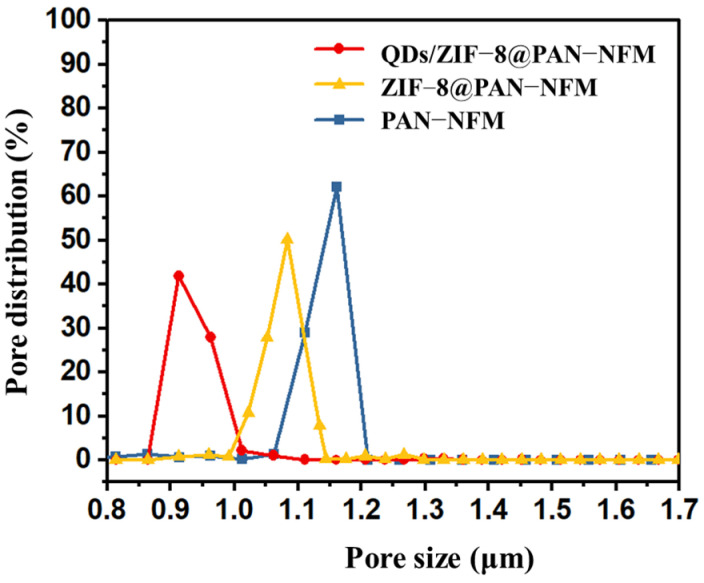
The pore size distribution of the PAN-NFM, ZIF-8@PAN-NFM, and QDs/ZIF-8@PAN-NFM.

**Figure 6 ijms-25-10346-f006:**
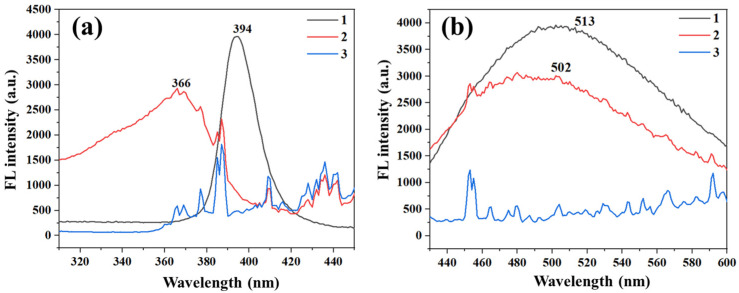
The fluorescence properties, (**a**) excitation spectra and (**b**) emission spectra, of the QD solution (1), QDs/ZIF-8@PAN-NFM (2), and ZIF-8@PAN-NFM (3).

**Figure 7 ijms-25-10346-f007:**
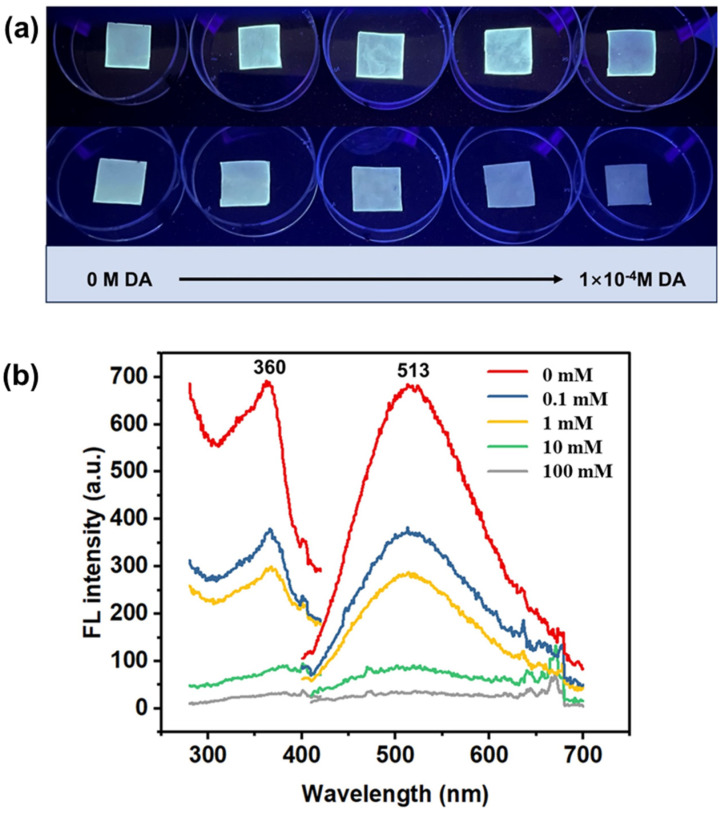
(**a**) Pictures of the QDs/ZIF-8@PAN-NFM (upper) and the QDs/ZIF-8@PAN-NFM (lower) under UV light and applied with different concentrations of DA. (**b**) Merged spectra of the fluorescence excitation (left) and emission (right) of the QDs/ZIF-8@PAN-NFM.

**Figure 8 ijms-25-10346-f008:**
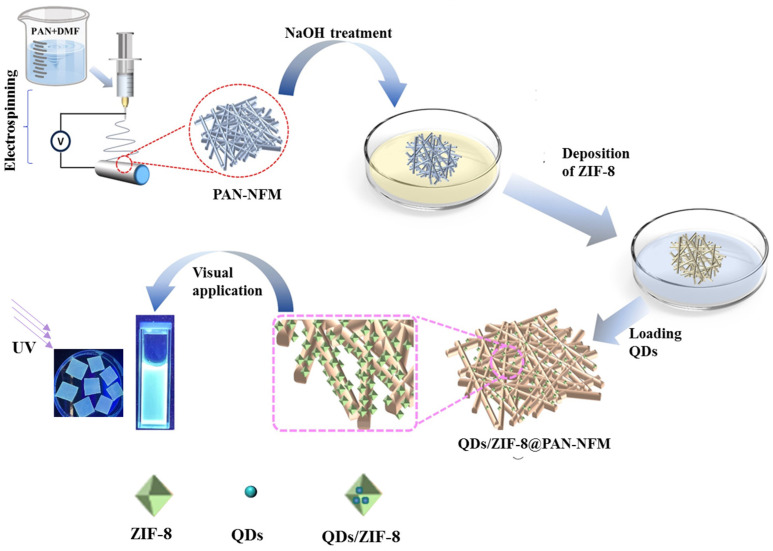
An illustration of the preparation process of adsorbing CdS/ZnS QDs onto the ZIF-8@PAN-NFM for visually testing DA.

**Figure 9 ijms-25-10346-f009:**
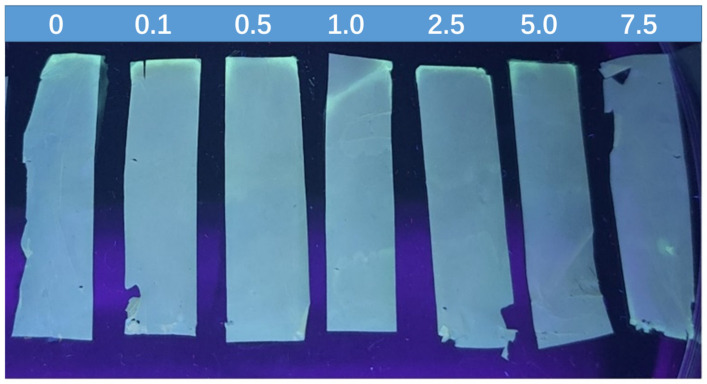
The fluorescence intensity under UV light of the QDs/ZIF-8@PAN-NFM with DA concentrations (0, 0.1, 0.5, 1.0, 2.5, 5.0, 7.5 mmol/L).

**Figure 10 ijms-25-10346-f010:**
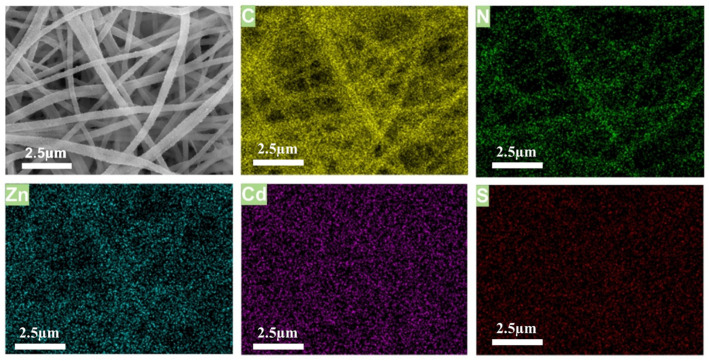
The SEM and EDS spectra images of the QDs/ZIF-8@PAN-NFM.

## Data Availability

Data are contained within the article.

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
