# Peer review of "Visual Detection of Dopamine with CdS/ZnS Quantum Dots Bearing by ZIF-8 and Nanofiber Membranes"

_ijms, 2024, doi:10.3390/ijms251910346_

Round 1
Reviewer 1 Report
Comments and Suggestions for Authors
The manuscript prepared QDs/ZIF-8@PAN-NFM material, and dopamine was detected with sensitive and visual, but there are some issues that the author should consider:
1. The TEM image of the material should be provided.
2. What does the vertical axis in Figure 9 represent?
3. From Figure 9b, it can be seen that compared to quantum dots, the fluorescence spectrum of QDs/ZIF-8@PAN-NFM not only shifts blue but also a decrease in fluorescence intensity. What is the significance of detecting DA?
4. Why are there two broken spectra at the same concentration in Figure 10?
5. My fifth point means that in order to be consistent with the author's title, it is necessary to conduct a comprehensive analysis and detection. That is, to establish methodology for dopamine determination based on the material the author prepared. Including linear range, linear equation, detection limit, etc., which the author did not provide. If the author simply wants to express the development of a new material and see if it has any practical value. There is no problem using dopamine for simple analytical applications as in the current manuscript. In this case, the author's current title cannot be used. That's what I meant by point 5.
Comments on the Quality of English LanguageEnglish writing needs improvement.
Reviewer 2 Report
Comments and Suggestions for Authors
1. On page 4, line 142, In figures 2(d) & (e), shows many particles attached to the surface of fibers, is there any effect on the QD and performance? The authors should explain about it.
2. On page 5, lines 157 to 159, Figure 4 QDs/ZIF-8@PAN-NFM SEM and EDS spectra should be Figure 3 instead of Figure 4.
3. In Figure 5, the Author mentions that 1590 cm−1 and 1145 cm−1 are attributed to ZIF-8. How about the 2929, 1145 and 758 cm−1? Also, what's the wavelength of QDs?
4. In Figure 6, why do QDs/ZIF-8 have the highest weight loss from 100 to 40%?
5. On page 6, Line 178, the Author mentions that the weight losses of ZIF8@PAN-NFM and QDs/ZIF-8@PAN-NFM were 48% and 57%, respectively. What are the temperature conditions?
6. On page 7, Line 202, “High porosity facilitates the diffusion of QDs solution to ZIF-8, improving the response rate. After growing ZIF-8 in situ, the pore size of ZIF-8@PAN-NFM decreases, ranging from 1.0 -1.15 µm, with a pore size of 1.08 µm accounting for 50.0%”. So based on those conditions, the porosity of QD in ZIF-8@PAN-NFM should be lower than PAN-NFM right? What's the purpose of the ZIF-8?
7. In Figure 9, What is the emission mode of the light source and wavelength you use? Why the wavelength of excitation and emission in the x-axis is not the same?
8. In the excitation mode of QD-S, the FWHM looks narrower than QD-ZIF-8@PAN, any reason?
9. Figure 10(a) down shows the ZIF-8@PAN-NFM without DA, as mentioned in the figure caption, so what are the conditions from left to right?
10. In Figure 10(b), why the wavelength is not continuous at 400 nm? In the fluorescence, the wavelength of the excitation UV source is 360 nm or not? If so, why there is no UV FL intensity in 10mM & 100mM?
11. In Figure 11, there are 7 conditions of DA, but shows 8 samples, any reasons?
